# GSK3β Inhibition Is the Molecular Pivot That Underlies the Mir-210-Induced Attenuation of Intrinsic Apoptosis Cascade during Hypoxia

**DOI:** 10.3390/ijms23169375

**Published:** 2022-08-19

**Authors:** Gurdeep Marwarha, Øystein Røsand, Katrine Hordnes Slagsvold, Morten Andre Høydal

**Affiliations:** 1Group of Molecular and Cellular Cardiology, Department of Circulation and Medical Imaging, Faculty of Medicine and Health, Norwegian University of Science and Technology (NTNU), 7034 Trondheim, Norway; 2Department of Cardiothoracic Surgery, St. Olavs University Hospital, 7030 Trondheim, Norway

**Keywords:** miR-210, hypoxia, AC-16 cardiomyocytes, apoptosis, GSK3β, BAX, BAK, Cytochrome C, mitochondria

## Abstract

Apoptotic cell death is a deleterious consequence of hypoxia-induced cellular stress. The master *hypoxamiR*, microRNA-210 (miR-210), is considered the primary driver of the cellular response to hypoxia stress. We have recently demonstrated that miR-210 attenuates hypoxia-induced apoptotic cell death. In this paper, we unveil that the miR-210-induced inhibition of the serine/threonine kinase Glycogen Synthase Kinase 3 beta (GSK3β) in AC-16 cardiomyocytes subjected to hypoxia stress underlies the salutary protective response of miR-210 in mitigating the hypoxia-induced apoptotic cell death. Using transient overexpression vectors to augment miR-210 expression concomitant with the ectopic expression of the *constitutive active* GSK3β S9A mutant (*ca*-GSK3β S9A), we exhaustively performed biochemical and molecular assays to determine the status of the hypoxia-induced intrinsic apoptosis cascade. Caspase-3 activity analysis coupled with DNA fragmentation assays cogently demonstrate that the inhibition of GSK3β kinase activity underlies the miR-210-induced attenuation in the hypoxia-driven apoptotic cell death. Further elucidation and delineation of the upstream cellular events unveiled an indispensable role of the inhibition of GSK3β kinase activity in mediating the miR-210-induced mitigation of the hypoxia-driven BAX and BAK insertion into the *outer mitochondria membrane* (*OMM*) and the ensuing *Cytochrome C* release into the cytosol. Our study is the first to unveil that the inhibition of GSK3β kinase activity is indispensable in mediating the miR-210-orchestrated protective cellular response to hypoxia-induced apoptotic cell death.

## 1. Introduction

Apoptosis is a significant contributor and a major conduit for the cardiomyocyte cell death observed in acute myocardial infarction (AMI) [1,2,3,4] and the post-AMI-manifested left ventricular dysfunction [5] that culminates in post-ischemic dilated cardiomyopathy [2,6]. Ischemic heart disease (IHD), the underlying etiological factor that could manifest clinically as AMI, is characterized by chronic persistent myocardial ischemia and cellular hypoxia known to activate both the *extrinsic* apoptotic and *intrinsic* apoptotic pathways. Chronic myocardial ischemia leading to cellular hypoxia evokes significant changes in the cardiomyocyte *transcriptome* [7,8], *proteome* [7,9,10], and *signalosome* [11,12], resulting in primary as well as ensuing concatenated gene expression changes [13,14,15] that are largely attributed to the fluxes in microRNA (miR) expression [16,17,18]. The hypoxia-induced expression of the *hypoxamiR* (hypoxia-induced microRNA), microRNA-210 (miR-210), is an indispensable molecular hallmark of the (mal)adaptive cellular response to cellular hypoxia [19,20,21,22,23,24]. In the last decade, a multitude of clinical and laboratory studies have implicated miR-210 as the most significant modulator of the hypoxia-induced derangements in mitochondrial function [25,26], as well as the modulation of the ensuing apoptotic cell death [27,28,29,30,31,32,33,34]. We have recently shown that miR-210 attenuates the hypoxia-induced activation of the intrinsic apoptosis pathway [35]. Despite miR-210 being implicated as the indispensable molecular component of the orchestrated cellular adaptive response that modulates apoptotic cell death in response to hypoxia, the proximal and distal molecular effectors that mediate this cellular adaptive response to hypoxia, remain elusive and egregiously characterized. The evolutionary conserved serine/threonine kinase, Glycogen Synthase Kinase 3 beta (GSK3β), is considered a pivotal signaling node that integrates upstream converging signaling pathways and modulates downstream diverging pathways that critically regulate cell survival and cell death [36,37]. GSK3β activation promotes the intrinsic apoptosis pathway in response to several noxious stimuli [36,37], through augmenting mitochondrial outer membrane permeabilization, in a multitude of rodent models [38,39,40] and immortalized cell lines [41,42]. The inhibition of GSK3β activity, as a consequence of phosphorylation of the Ser^9^ residue [43], confers a cardioprotective response in a multitude of rodent models of myocardial ischemia and AMI [38,40,44,45,46,47,48]. The inhibition of GSK3β activity is principally effectuated by Akt, the most proximal and hegemonial kinase that phosphorylates GSK3β at the Ser^9^ residue [49]. Emerging evidence implicates miR-210 in the regulation of upstream molecular components of the Akt signaling pathway in immortalized cancer cell-lines [50,51], although the studies did not directly determine the miR-210-regulation of Akt activity. However, to date, no studies have determined the impact of miR-210 on the kinase activity of GSK3β, a well-characterized Akt substrate and an indispensable molecular entity in the modulation of hypoxia-induced apoptotic cell death. In this study, we determined the impact of miR-210 on the kinase activity of GSK3β and exhaustively delineated the role of GSK3β (in)activation in effectuating the miR-210-induced attenuation in apoptotic death.

## 2. Results

### 2.1. miR-210 Attenuates the Hypoxia-Induced GSK3β Activation

To elucidate and dissect the molecular signaling mechanisms that could underlie the effects of miR-210 on apoptotic cell death during hypoxia, we first determined the effects of miR-210 overexpression on the hypoxia-induced GSK3β activation. The evolutionary conserved serine/threonine protein kinase GSK3β is one of the most significant intracellular effector and modulator of the *intrinsic* apoptotic cascade [36,52,53]. GSK3β exhibits constitutive and robust basal intrinsic kinase activity that is molecularly governed by phosphorylation of the Ser^9^ residue that results in its inactivation and loss of the basal intrinsic kinase activity [54], as well as by phosphorylation of the Tyr^216^ residue within the activation loop that enhances the basal intrinsic kinase activity [55]. Using quantitative densitometry-coupled Western blot analysis, we determined the levels of p-Ser^9^ GSK3β and p-Tyr^216^ GSK3β as a surrogate measure of the activation status of GSK3β. We found that hypoxia challenge elicited a pronounced decrease in the inhibitory phosphorylation at the Ser^9^ residue concomitant with no changes in the activating phosphorylation at the Tyr^216^ residue (Figure 1A–C). Overexpression of miR-210 significantly attenuated the hypoxia-induced decrease in the inhibitory phosphorylation at the Ser^9^ residue (Figure 1A,B) concomitant with eliciting no changes in the hypoxia-unresponsive activating phosphorylation at the Tyr^216^ residue (Figure 1A,C). To further corroborate and demonstrate the mitigative effects of miR-210 on the hypoxia-induced GSK3β activation, we subsequently determined direct GSK3β kinase activity [56,57] in non-denatured lysates. The effects of miR-210 modulation on GSK3β kinase activity were consistent with those observed with the effects of miR-210 modulation on the phosphorylation status of GSK3β at the Ser^9^ residue. Hypoxia challenge evoked a pronounced increase in the kinase activity of GSK3β that was significantly attenuated by the overexpression of miR-210 (Figure 1D). The validity of miR-210 overexpression in the respective cell lysates was determined by miR-210 hybridization immunoassay (Figure 1D). Taken together, the data from GSK3β kinase activity assay and quantitative densitometry-coupled Western blot analysis determining the levels of p-Ser^9^ GSK3β demonstrate that miR-210 significantly attenuates the hypoxia-induced GSK3β activation.

### 2.2. GSK3β Inhibition Is a Necessary Molecular Event for the miR-210-Induced Attenuation of Apoptotic Cell Death during Hypoxia

To further delineate and substantiate the role of GSK3β inactivation in the miR-210-mediated mitigation of apoptotic cell death, we performed biochemical assays to determine hypoxia-induced apoptotic cell death in AC-16 cells ectopically expressing the GSK3β-S9A mutant that results in *constitutively active* GSK3β (*ca*-GSK3β-S9A) [58]. When LDH (lactate dehydrogenase) released into the conditioned media was determined as a surrogate measure of generic cell death, ectopic expression of the *ca*-GSK3β-S9A mutant significantly abrogated the endogenous miR-210 overexpression-induced reduction of the hypoxia-evoked increase in LDH release (Figure 2A). To further specifically determine whether the miR-210-evoked GSK3β inactivation underlies the miR-210-mediated mitigation of apoptotic cell death during hypoxia challenge, we determined caspase-3 activation in AC-16 cells ectopically expressing the *ca*-GSK3β-S9A mutant concomitant with the miR-210 overexpression, in the context of hypoxia challenge. To this end, using Western blot analysis and sandwich ELISA immunoassay, we first determined the expression levels of the active *cleaved*-caspase-3 (p17 fragment) relative to the inert procaspase-3 (p32) expression levels as a surrogate marker of caspase-3 activation status (Figure 2B). Overexpression of the endogenous miR-210 significantly mitigated the hypoxia-induced increase in the proteolytic processing of procaspase-3 (p32) into *cleaved*-caspase-3 (p17 fragment) as visualized by Western blot analysis (Figure 2B) and quantitatively determined by sandwich ELISA immunoassay (Figure 2C). However, ectopic overexpression of the *ca*-GSK3β-S9A mutant significantly abrogated this miR-210-elicited reduction in the hypoxia-induced increase in the proteolytic processing of procaspase-3 (p32) into *cleaved*-caspase-3 (p17 fragment) (Figure 2B,C). Caspase-3 activation results in the cleavage and inactivation of the DNA repair enzyme PARP (Poly [ADP-ribose] polymerase), a bonafide substrate of caspase-3 [59,60]. Furthermore, the caspase-3 mediated cleavage of PARP is widely considered a molecular hallmark of early apoptosis [59,61,62]. Ergo, we further determined caspase-3 activation status by determining the expression levels of the inactivated *cleaved*-PARP (p89 fragment) relative to the expression levels of full-length active PARP (p116). Overexpression of the endogenous miR-210 significantly attenuated the hypoxia-induced increase in the caspase-3-mediated proteolytic processing of p116 PARP into the p89 *cleaved*-PARP fragment as determined by Western blot analysis (Figure 2D) and quantitative sandwich ELISA immunoassay (Figure 2E). Ectopic overexpression of the *ca*-GSK3β-S9A mutant significantly ablated this miR-210-elicited attenuation in the hypoxia-induced increase in the caspase-3-mediated proteolytic processing of p116 PARP into the p89 *cleaved*-PARP fragment (Figure 2D,E). We subsequently determined the miR-210 overexpression-induced attenuation in the hypoxia-evoked caspase-3 activity in AC-16 cells that ectopically co-express the *ca*-GSK3β mutant (Figure 2F). The miR-210 overexpression-induced attenuation in the hypoxia-evoked caspase-3 activity was significantly ablated in AC-16 cells ectopically expressing the *ca*-GSK3β mutant (Figure 2F). miR-210 overexpression in the respective cellular lysates was validated by the miR-210 hybridization immunoassay (as described in Section 4.3) and are reported in Appendix A. The ectopic expression of the HA-tagged *ca*-GSK3β-S9A mutant was validated by Western blot analysis (Figure 2B) as well as by ELISA immunoassay (Appendix A). GSK3β kinase activity (Appendix A) was determined in the entire gamut of *native* lysates to corroborate and validate the translative effects of the ectopic expression of the *ca*-GSK3β-S9A mutant.

We next determined DNA fragmentation, a cardinal morphological hallmark of late-stage apoptosis [63,64], using the terminal deoxynucleotidyl transferase dUTP nick end labeling (TUNEL) method [65]. The miR-210-elicited reduction in the hypoxia-induced DNA fragmentation was significantly ablated in AC-16 cells ectopically expressing the *ca*-GSK3β-S9A mutant (Figure 3A). We subsequently determined the extent of cellular markers and molecular correlates of DNA fragmentation [66]. The endonuclease *DNA fragmentation factor subunit beta (DFFβ/DFF40)*, also called *Caspase-activated DNase (CAD),* is considered the primary molecular instigator of apoptotic DNA fragmentation [66,67,68]. DFF40 (DFFβ, CAD) is sequestered in an *inert* heterodimer with its cognate protein, *DNA fragmentation factor subunit alpha (DFFα/DFF45)*, also called *Inhibitor of Caspase-activated DNase (ICAD)* [66,67,68]. The caspase-3 induced cleavage of DFF45 (DFFα, ICAD) results in the dissociation of DFF45 (DFFα, ICAD) from DFF40 (DFFβ, CAD) and the consequent de-repression of the DFF40 (DFFβ, CAD) endonuclease activity [66,67,68]. We therefore determined the caspase-3-induced proteolytic cleavage of DFF45 (DFFα, ICAD) as a molecular correlate of DNA fragmentation [66,67,68]. Overexpression of miR-210 significantly ablated the hypoxia-induced cleavage of DFF45 (DFFα, ICAD) into the p11 fragment (Figure 3B,C). However, this miR-210-elicited ablation of the hypoxia-induced proteolytic processing of DFF45 (DFFα, ICAD) into the p11 fragment was not observed in AC-16 cells ectopically expressing the *ca*-GSK3β-S9A mutant (Figure 3B,C). We subsequently identified the endonuclease activity of DFF40 (DFFβ, CAD) as a direct molecular correlate of DNA fragmentation. The hypoxia-induced DFF40 (DFFβ, CAD) endonuclease activity was significantly attenuated in AC-16 cells overexpressing miR-210 (Figure 3D). However, ectopic overexpression of the *ca*-GSK3β-S9A mutant concomitant with miR-210 overexpression resulted in the complete loss of the miR-210-elicited mitigation of hypoxia-induced DFF40 (DFFβ, CAD) endonuclease activity (Figure 3D). Taken together, these data demonstrate that the miR-210-elicited reduction in hypoxia-evoked DNA fragmentation and apoptotic cell death is significantly mediated through the inhibition of GSK3β kinase activity.

### 2.3. miR-210 Mitigates the Hypoxia-Induced Intrinsic Apoptotic Pathway through GSK3β Inhibition

Having established that miR-210 attenuates apoptotic cell death through the inhibition of GSK3β kinase activity, we further determined the upstream molecular cues and events that culminate in caspase-3 activation and apoptotic cell death. Our previous study has cogently demonstrated that miR-210 mitigates the activation of the *intrinsic apoptotic pathway* that entails *Cytochrome C* release leading to the *apoptosome* formation. In this study, we therefore sought to determine whether miR-210 mitigates the activation of the *intrinsic apoptotic pathway* through the inhibition of GSK3β kinase activity. To this end, we isolated and segregated the mitochondrial and cytosolic compartments by cellular fractionation (Section 4.11) The integrity of the mitochondrial fraction was validated by the presence of COX4 (Cytochrome C Oxidase Subunit 4) concomitant with the absence of β-Actin, while the corollary criteria, i.e., absence of COX4 concomitant with the presence of β-Actin, was used to validate the integrity of the cytosolic fraction (Appendix A). We determined *Cytochrome C* translocation from the mitochondria into the cytosol as a surrogate measure of the activation status of the *intrinsic apoptotic pathway*. In cells ectopically expressing the *ca*-GSK3β-S9A mutant, overexpression of miR-210 failed to elicit a significant mitigation in the hypoxia-induced release of *Cytochrome C* from the mitochondria into the cytosol, as unveiled by sandwich ELISA immunoassays determining *Cytochrome C* levels in the mitochondrial fraction (Figure 4A) relative to the cytosolic fraction (Figure 4B). We further characterized the effects of miR-210-induced inhibition of GSK3β kinase activity on the hypoxia-driven upstream molecular entities that govern mitochondrial permeability transition and the release of *Cytochrome C* into the cytosol leading to the activation of the intrinsic apoptosis *pathway*. To this end, we first determined the expression levels and the activation status of the BCL2 (B cell lymphoma 2) family of proteins, which dictate mitochondrial permeability transition through the formation of the high-conductance pore, *MAC* (mitochondrial apoptosis-induced channel) [69,70,71], leading to an increase in the mitochondrial outer membrane permeabilization. The multiple BH (Bcl2 Homology) domain-containing pro-apoptotic proteins, BAX (BCL2-associated X) and BAK (BCL2-homologous antagonist/killer), are the terminal effectors and structural molecular components of the *MAC pore complex* in the *outer mitochondrial membrane* (*OMM*) [72,73]. The BAX-/BAK-induced formation of the *MAC pore complex* in the *OMM*, is preceded by the translocation of BAX (but not BAK as it is constitutively localized to the mitochondria) from the cytosol to the mitochondria [74,75], followed by conformational changes in their structure and *OMM* anchoring that eventually leads to the formation of higher-order oligomers that insert into the *OMM* forming the *MAC* pore [73,76]. We therefore determined the relative abundances of BAX and BAK in the mitochondria versus the cytosol. Quantitative sandwich ELISA immunoassays, performed in mitochondrial fractions and cytosolic fractions, revealed that the overexpression of miR-210 significantly mitigates the hypoxia-induced translocation and enrichment of BAX (Figure 4C,D), but not BAK (Figure 4E,F), into the mitochondria. However, in cells ectopically expressing the *ca*-GSK3β S9A mutant, the miR-210-elicited mitigation of the hypoxia-induced translocation of BAX into the mitochondria was abrogated (Figure 4C,D). No changes in the total BAX and BAK levels were observed in the whole-cell lysates across the entire gamut of experimental groups (Appendix A), suggesting that the expression level changes did not underlie the aforementioned observed responses.

The efficacy of BAX and BAK in eliciting mitochondrial outer membrane permeabilization, is negatively regulated by their sequestration by members of the anti-apoptotic BCL2 family of proteins, namely BCL2, BCL-X_L_, and MCL1. The molecular sequestration of BAX and BAK in a heterodimeric complex with BCL-X_L_, BCL2, and MCL1 results in their retro-translocation from the *OMM* to the cytosol and therefore titrates their abundance in the *OMM*. We therefore determined the relative abundances of BCL2, BCL-X_L_, and MCL1 in the mitochondrial fractions versus the cytosolic fraction, as fluxes in BCL2, BCL-X_L_, and MCL1 levels in the mitochondria could confer an increase in BAX and BAK localization in the *OMM*, fostering their oligomerization that culminates in the formation of the *MAC* pore. Quantitative sandwich ELISA immunoassays unveiled that neither the miR-210 overexpression nor the ectopic expression of the *ca*-GSK3β S9A mutant, elicited any change in the hypoxia-induced decrease in BCL-X_L_ levels in the mitochondrial compartment relative to the cytosolic compartment (Figure 5A,B). The hypoxia-induced retro-translocation of BCL2 from the mitochondria to the cytosol was also not affected by miR-210 overexpression, with the ectopic expression of the *ca*-GSK3β S9A mutant exacerbating the hypoxia-induced retro-translocation of BCL2 from the mitochondria to the cytosol (Figure 5C,D). No changes in the total BCL-X_L_ and BCL2 levels were observed in the whole-cell lysates across the entire gamut of experimental groups (Appendix A), suggesting that the expression-level changes did not underlie the aforementioned observed responses. Interestingly, hypoxia induced a profound decrease in MCL1 levels in all the three determined cellular fractions, namely the mitochondrial fraction (Figure 5E), the cytosolic fraction (Figure 5F), and the whole-cell lysates (Appendix A). Furthermore, miR-210 attenuated the hypoxia-induced decrease in MCL1 levels in all three determined aforementioned cellular fractions (Figure 5E,F and Appendix A). However, the miR-210-elicited mitigation in the hypoxia-induced decrease in MCL1 levels in all three determined cellular fractions was abrogated in cells ectopically expressing the *ca*-GSK3β S9A mutant (Figure 5E,F and Appendix A). Taken together, these data implicate the miR-210-evoked inhibition of hypoxia-driven GSK3β kinase activity as an indispensable molecular conduit for the miR-210-elicited attenuation of the hypoxia-driven BAX translocation to the mitochondria concomitant with the preclusion of the hypoxia-induced decrease in MCL1 expression levels.

The abundance of BAX and BAK in the mitochondrial fractions could be ascribed to two different pools: one that is constituted by BAX and BAK monomers that are tethered to the *OMM* [73] and the other that is constituted by the higher-order BAX and BAK oligomers that insert onto the *OMM* as integral membrane proteins, thereby forming the *MAC* [73]. We aimed at distinguishing the aforementioned respective pools to quantitatively determine the abundance of higher-order BAX and BAK oligomers that insert into the *OMM* as integral membrane proteins.

To this end, we fractionated the isolated mitochondria into the *OMM-inserted* (*OMM*-embedded) and the *OMM-tethered* (*OMM*-anchored) protein fractions [77,78,79] (Section 4.14) followed by the determination of BAX and BAK abundance in the respective fractions. We first performed ELISA immunoassays for the bonafide mitochondrial-resident proteins to establish the integrity and validity of the *OMM-inserted* and *OMM-tethered* protein fractions. The integrity of the *OMM-inserted* protein fraction was validated by the presence of TOM40 and TIM22 concomitant with the absence of SDHA and HK2 (Appendix A). In the corollary analysis, the integrity of the *OMM-tethered* protein fraction was validated by the absence of TOM40 and TIM22 concomitant with the presence of SDHA and HK2 (Appendix A). Quantitative sandwich ELISA immunoassays for BAX and BAK in the respective mitochondrial membrane fractions revealed that miR-210 overexpression elicited a profound mitigation in the hypoxia-induced increase in the abundance of BAX (Figure 6A) and BAK (Figure 6B) in the *OMM-inserted* protein fraction. However, this miR-210-elicited attenuation of the hypoxia-induced increase in the abundance of *OMM-inserted* BAX (Figure 6A) and BAK (Figure 6B) was not observed in cells that concomitantly expressed the *ca*-GSK3β S9A mutant, thereby implicating the inhibition of GSK3β kinase activity as the underlying molecular conduit. Further analysis of BAX in the *OMM-tethered* protein fraction revealed analogous observations to those seen in the *OMM-inserted* protein fraction, whereby miR-210 overexpression elicited a profound attenuation in the hypoxia-induced increase in the abundance of *OMM-tethered* BAX (Figure 6C) that was contingent on the inhibition of GSK3β kinase activity. Apropos of BAK analysis in the *OMM-tethered* protein fraction, quantitative sandwich ELISA immunoassays unveiled that hypoxia induces a significant decrease in the abundance of the basally resident *OMM-tethered* BAK (Figure 6D) that contributes to a commensurate increase in the profound hypoxia-induced increase in *OMM-inserted* BAK (Figure 6B). Furthermore, miR-210 overexpression significantly attenuated the hypoxia-induced decrease in *OMM-tethered* BAK (Figure 6D) concomitant with an attenuation in the hypoxia-induced increase in the *OMM-inserted* pool of BAK (Figure 6B), which was contingent on the inhibition of GSK3β kinase activity. Collectively, these data demonstrate that miR-210 attenuates the hypoxia-driven increase in the abundance of *OMM-inserted* BAX and BAK through the inhibition of GSK3β kinase activity. In an integrated overview, our data suggest that the inhibition of GSK3β kinase activity underlies the miR-210 elicited attenuation in the hypoxia-driven increase in the OMM-insertion of BAX and BAK oligomers, the consequential release of Cytochrome C into the cytosol, and the ensuing caspase-3 and DFF40 endonuclease activation leading to apoptotic cell death through the inhibition of GSK3β. The overall inferences drawn from our data, representing the findings and observations that the inhibition of GSK3β kinase activity is an indispensable molecular event and mediator of the miR-210-elicited mitigation of hypoxia-induced apoptotic cell death, are summated and presented as an illustrative schematic model in Figure 7.

## 3. Discussion

The novel finding that miR-210 attenuates hypoxia-induced apoptotic cell death through the inhibition of GSK3β kinase activity bears huge implications in the realm of our comprehension of the molecular entities that are intricately involved in cellular hypoxia and tissue-ischemia-evoked pathogenesis, which is observed in post-AMI associated left ventricular dysfunction and dilated cardiomyopathy. Firstly, we identified GSK3β as a distal molecular target of miR-210, albeit an indirect target, as we found that miR-210 augments the inhibitory phosphorylation of GSK3β at the Ser^9^ residue and consequently mitigates the kinase activity of GSK3β without affecting the expression levels. Secondly, we identified that this miR-210-elicited reduction of GSK3β kinase activity is indispensable for the miR-210-induced attenuation of hypoxia-induced cardiomyocyte cell death. These are seminal and profound observations, as despite overwhelming evidence from clinical and laboratory studies implicating miR-210 as an indispensable molecular component of the (mal)adaptive cellular response to hypoxia, the underlying downstream effectors and molecular entities mediating the miR-210 orchestrated cellular response have largely remained elusive and undefined to date.

The canonical role of GSK3β in mediating apoptotic cell death in response to growth factor withdrawal and/or other noxious stimuli, including cellular hypoxia and ischemic stress, has been expounded on for two decades and evinced by innumerable laboratory and clinical studies [36]. The novelty of the findings reported in our study further confer GSK3β signaling as a distal hub of the signaling cross-talk fluxes that are targeted by the master *hypoxamiR*, miR-210, in the orchestrated cellular response to hypoxia stress. Furthermore, we identified GSK3β as a constituent of the miR-210 *regulome,* and not the miR-210 *targetome,* as our study did not observe any miR-210-mediated direct modulation of GSK3β protein levels. Rather, our findings unveil miR-210 as a distal regulator of GSK3, which impinges upon the upstream (in)activators that modulate GSK3β kinase activity. Therefore, questions are still abound pertaining to the identity of the miR-210-regulated upstream (in)activators and downstream proximal effectors of GSK3β kinase activity during hypoxic stress. It must be noted, apropos of the upstream (in)activators of GSK3β kinase activity, there is some empirical evidence that miR-210 modulates Akt activity [50,51], the most proximal and hegemonic regulator of GSK3β kinase activity [49,80,81]. Further studies are therefore warranted to delve into the miR-210-mediated regulation of Akt kinase activity and subsequently dissect its role in the miR-210-elicited inhibition of GSK3β kinase activity and the ensuing mitigation in hypoxia-induced activation of the intrinsic apoptosis pathway. Despite the monolithic hegemony exhibited by Akt in regulating GSK3β kinase activity through Ser^9^ phosphorylation, it must be emphasized that a multitude of protein kinases belonging to the AGC family of protein kinases—including PKA (Protein Kinase A), p70S6K2 (70 kDa Ribosomal Protein S6 Kinase 2), p90RSK1 (90 kDa Ribosomal Protein S6 Kinase 1), and PKCα (Protein Kinase C Alpha)—are known to inhibit GSK3β kinase activity through Ser^9^ phosphorylation [82]. Thus, from a molecular mechanistic perspective, further studies encompassing a panoramic approach determining the effects of miR-210 on a multitude of kinase *signalosomes* is warranted, to completely delineate the upstream molecular signaling components that mediate the inhibitory effects of miR-210 on GSK3β kinase activity.

Enhanced miR-210 expression has been proposed as a viable molecular strategy to combat ischemic tissue damage [83], augment cardiomyocyte survival [29], and foster cardiac repair [84] and angiogenesis post-MI [29]. However, the molecular effectors that mediate the cellular cardioprotective effects of miR-210 are not delineated and, at best, are egregiously comprehended. Our seminal finding reported in this study, that miR-210 attenuates hypoxia-driven intrinsic apoptosis by inhibiting hypoxia-driven GSK3β kinase activity, thus bears significant clinical implications as it unveils GSK3β as a novel molecular target of miR-210. The last two decades have witnessed an emerging consensus that the inhibition of GSK3β kinase activity could be a putative pharmacological target in myocardial ischemia and post-MI [44,85]. A multitude of laboratory studies have implicated the inhibitory phosphorylation of GSK3β at Ser^9^ as a requisite molecular event for the ischemic preconditioning-elicited cardioprotective response [38,40,45,48]. Furthermore, the ischemic postconditioning-elicited reduction in infarct size in an in vivo rodent model of acute myocardial ischemia-reperfusion injury also implicates the inhibitory phosphorylation of GSK3β at Ser^9^ as the molecular basis for the ischemic postconditioning-elicited cardioprotective response [40]. Thus, in the context of cardioprotective strategies, our finding that miR-210 inhibits GSK3β activity through augmenting the Ser^9^ phosphorylation unveils a unique dimension of miR-210 biology and opens a new frontier and research avenue in determining the full spectrum of the therapeutic potential of miR-210.

In a broader cellular and biochemical context, our finding that miR-210 attenuates intrinsic apoptotic cell death through the inhibition of GSK3β, and the prevailing fundamental tenets that have established the consensus of a proapoptotic disposition of GSK3β, must be critically examined and expounded on further. A plethora of laboratory studies executed in a diverse array of disease model systems and experimental paradigms, have concluded a *bi-functional paradoxical* role of GSK3β in regulating apoptotic cell death [36,86,87]. A complex tapestry of the proapoptotic and antiapoptotic roles of GSK3β has emerged, whereby overwhelming evidence implicates GSK3β activation in effectuating the mitochondria-evoked intrinsic apoptosis pathway, while attenuating the death receptor-induced *extrinsic apoptosis* pathway [36]. Interestingly, we have recently demonstrated that miR-210 modulates the hypoxia-induced apoptotic cell death in a diametrically opposite manner, whereby miR-210 mitigates the hypoxia-induced intrinsic apoptosis pathway while exacerbating the hypoxia-induced activation of the death receptor-induced *extrinsic apoptosis* pathway [35]. Thus, it is in agreement that further studies should be designed to determine the role of GSK3β (in)activation in the miR-210-induced exacerbation of the hypoxia-induced activation of the death receptor-induced *extrinsic apoptosis* pathway.

## 4. Materials and Methods

### 4.1. Cell Culture and Treatments

Human AC-16 cardiomyocyte cells (EMD Millipore/Merck Millipore/Merck Life Sciences, Catalogue # SCC109, Darmstadt, Germany, RRID:CVCL_4U18) were cultured and sub-cultured in the standard maintenance medium, Dulbecco’s modified Eagle’s medium (DMEM): Ham’s F12 (1:1; *v*/*v*) with 2 mM Glutamine, 12.5% fetal bovine serum (FBS), and 1% antibiotic/antimycotic mix, in accordance with the standard guidelines, procedures, and protocols established by the commercial vendor. AC-16 cells were *reverse*-transfected with either the *pEZX-MR04-miR-210* expression vector (GeneCopoeia^TM^, Rockville, MD, USA, Catalogue # HmiR0167-MR04) or the corresponding *pEZX-MR04-scrambled* expression vector (GeneCopoeia^TM^, Rockville, MD, USA, Catalogue # CmiR0001-MR04) concomitant with either the *pcDNA3-HA-GSK3β S9A* expression vector (HA GSK3 beta S9A pcDNA3 vector was a gift from Jim Woodgett (Addgene plasmid # 14,754; http://n2t.net/addgene:14754, accessed on 15 June 2022); RRID:Addgene_14754) [58] or the corresponding pcDNA3 empty vector (Table 1), using Polyfect^®^ (Qiagen Norge, Oslo, Norway, Catalogue # 301107) in accordance to the manufacturer’s guidelines and standardized procedures. The plasmid load to be transfected was standardized to 1μg per 1.2 × 10^6^ cells and scaled up or scaled down in accordance with the stipulations of the experimental paradigm. The hypoxia challenge (1% O_2_, 5% CO_2_, and 94% N_2_ for 18 h) was effectuated by incubating the transfected AC-16 cells with the specific *hypoxia medium* (Appendix A) for 18 h. The experimental paradigm and experimental groups are defined in Table 1. Hypoxia challenge (1% O_2_, 5% CO_2_, and 94% N_2_ for 18 h) was induced and maintained for the designated duration using the *New Brunswick™ Galaxy^®^ 48 R CO_2_ incubator* (Eppendorf Norge AS, Oslo, Norway).

### 4.2. Western Blotting

Protein isolation from the whole-cell lysates was performed using standard protocols [88] and the protein concentration was determined with the Bradford protein assay. Briefly, treated AC-16 cells were washed 2× with PBS (phosphate-buffered saline), trypsinized to collect the cells, and centrifuged at 5000× *g*. The pellet was washed again 2× with PBS and homogenized in RIPA tissue lysis buffer (Tris 50 mM, sodium chloride (Nacl) 150 mM, Sodium Deoxycholate 0.5% *w*/*v*, sodium dodecyl sulphate (SDS) 0.1% *w*/*v*, Nonidet P-40 (NP-40) 1% *w*/*v*, pH 7.4) supplemented with protease and phosphatase inhibitors (Halt^TM^ Protease and Phosphatase Inhibitor Cocktail 100×, Thermo Fisher Scientific, Oslo, Norway, Catalogue # 78446). Mitochondrial and cytosolic fractions from the respective experimental groups were prepared using the mitochondrial fractionation kit, Cytochrome C Release Apoptosis Assay Kit, from Sigma Aldrich/Merck Millipore (Merck Life Science, Darmstadt, Germany, Catalogue # QIA87), following the manufacturer’s protocol and guidelines. Proteins (10–50 μg) were resolved on SDS-PAGE gels followed by transfer to a polyvinylidene difluoride (PVDF) membrane (Immun-Blot^TM^ PVDF Membrane, Bio-Rad Norway AS, Oslo, Norway, Catalogue # 1620177) and overnight incubation with the respective primary antibodies at 4 °C following standardized protocols [89]. The origin, source, dilutions of the respective antibodies used in this study are compiled in Table 2. β-actin was used as a gel loading control for whole-cell lysates and cytosolic fractions while COX4 was used as a gel loading control for the mitochondrial fractions. The blots were developed with enhanced chemiluminescence substrate (SuperSignal™ West Pico PLUS Chemiluminescent Substrate, Thermo Fisher Scientific, Oslo, Norway, Catalogue # 34580) and imaged using a LICOR Odyssey Fc imaging system (LI-COR Biotechnology, Cambridge, UK). Quantification of results was performed by densitometry using Image J (ImageJ, United States National Institutes of Health, Bethesda, MD, USA, https://imagej.nih.gov/ij/, accessed on 14 March 2021) [90] and the results were analyzed as total integrated densitometric values.

### 4.3. Enzyme-Coupled miR-210 Hybridization Immunoassay

The levels of miR-210 in the experimental cell lysates was determined by adopting a novel microRNA immunoassay approach [91,92]. This miR-210 immunoassay approach allowed the direct quantitative determination of miR-210 in the same experimental lysates being subjected to the specific downstream assays. Briefly, miR-210 in the experimental lysates was affinity-captured on streptavidin beads by hybridization with a biotin-labeled miR-210 *locked nucleic acid (LNA) capture probe* (Qiagen Norge, Oslo, Norway, Catalogue # 339,412 YCO0212944). The affinity-captured miR-210 was eluted from the streptavidin beads (10 mM Tris, pH 7.5 at 90 °C for 10 min), un-sequestered from the double-stranded hybrid by denaturation, and subsequently immobilized in the microwells of a *solid phase 96-well nucleic acid microplate* (Nunc™ NucleoLink™ Strips, Thermo Fisher Scientific, Oslo, Norway, Catalogue # 248259). The immobilized miR-210 was quantitated by adopting an indirect ELISA approach, whereby the immobilized miR-210 was hybridized with a digoxigenin-labeled miR-210 *LNA detection probe* (Qiagen Norge, Oslo, Norway, Catalogue # 339,412 YCO0212945) followed by immunodetection with the AP (alkaline phosphatase)-conjugated digoxigenin antibody (Digoxigenin AP-conjugated Antibody, R&D Systems, Minneapolis, MN, USA, Catalogue # APM7520) using the AP-substrate PNPP (p-Nitrophenyl Phosphate, disodium salt) (Thermo Fisher Scientific, Oslo, Norway, Catalogue # 37621) as the chromophore for the colorimetric read-out (λ_405_). Competition assays were also performed with the *unlabeled detection probe* to exhibit assay specificity and serve as an experimental blank. The raw optical density values measured at λ_05_ (405 nm) were corrected with the experimental blank and subsequently normalized and expressed as *fold-change* relative to the experimental control. Data are expressed as a *fold-change* ± standard deviation (S.D) from three technical replicates for each of the four biological replicates belonging to each experimental group, *n* = 4).

### 4.4. GSK3β Kinase Activity Assay

The GSK3β kinase activity in non-denatured experimental lysates was measured using an indirect ELISA immunoassay approach. The extent of phosphorylation of the serine residue in the synthetic peptide (RRRPASVPPSPSLSRHS*(pS)*HQRR) corresponding to the GSK3β consensus recognition motif in the endogenous substrate, muscle glycogen synthase 1 (where (pS) corresponds to phosphorylated serine), was used as a surrogate measure of GSK3β kinase activity in the experimental lysates. Next, 96-well microplates were prepared utilizing the specific streptavidin-biotin chemistry [93,94]. First, 96-well microplate wells were coated with 20 pmoles (1.1 μg) of streptavidin (110 μL of 10 μg/mL streptavidin in 20 mM potassium phosphate, pH 6.5), followed by immobilization of the biotinylated GSK3β substrate (GSM (GSK3 substrate peptide), Sigma Aldrich/Merck Life Science, Oslo, Norway, Catalogue # 12-533). The N-terminus biotinylation of the GSK3β substrate-peptide was performed using the “*EZ-Link™ NHS-LC-Biotin*” biotinylation kit (Thermo Fisher Scientific, Oslo, Norway, Catalogue # 21336) following the manufacturer’s guidelines and established protocols [95,96]. Briefly, 50 ng of the biotinylated GSK3β substrate-peptide was immobilized in each well of the streptavidin-coated 96-well microplate. The experimental cell lysates (equivalent to 50 μg of protein content) were incubated, with the immobilized biotinylated GSK3β substrate, overnight at 4 °C followed by incubation for 24 h with the specific detection-primary antibody (100 ng/well, 50 μL of 2 μg/mL) directed against the phosphorylated serine residue (Table 2). The abundance of the phosphorylated serine residue in the biotinylated GSK3β substrate was immunodetected with the AP-conjugated secondary antibody, using the AP-substrate PNPP as a chromophore for the colorimetric read-out (λ_405_). The GSK3β kinase activity assays were also performed in the presence of the GSK3β kinase inhibitor SB-216763 [97] (Sigma Aldrich/Merck Millipore/Merck Life Science, Darmstadt, Germany, Catalogue # S3442) to exhibit assay specificity and serve as an experimental blank. The respective optical density (O.D) values from the GSK3β kinase inhibitor-treated lysates were used for experimental blank correction. Data are expressed as experimental blank-corrected O.D_405_ (λ_405_) values from three technical replicates for each of the four biological replicates belonging to each experimental group (*n* = 4).

### 4.5. Lactate Dehydrogenase (LDH) Assay

The levels of lactate dehydrogenase (LDH) in the conditioned media were identified as a surrogate measure of generic cell death. LDH levels in the conditioned medium were measured using a sandwich ELISA immunoassay approach. Briefly, 20 ng of LDH capture antibody (Table 2) was immobilized in each well of a 96-well microplate [98]. The respective conditioned media (50 μL) from experimental samples were incubated with the immobilized LDH capture antibody overnight at 4 °C. The conditioned media was discarded, and the 96-well microplate wells were washed 3x (15 min each) with TBS-T (Tris-buffered saline with 0.1% *v*/*v* Tween-20) and incubated, with the LDH-A and LDH-B detection antibodies (Table 2), overnight at 4 °C. The 96-well microplate wells were washed 3× (15 min each) with TBS-T, followed by immunodetection with the AP-conjugated secondary antibody, using the AP-substrate PNPP as a chromophore for the colorimetric read-out (λ_405_). The antibodies and signal specificity were established by performing peptide blocking assays in the entire gamut of experimental lysates. The *LDH-A antibody blocking peptide* (Novus Biologicals/Bio-Techne, Abingdon, UK, Catalogue # NBP1-48336PEP) and *LDH-B antibody blocking peptide* (Novus Biologicals/Bio-Techne, Abingdon, UK, Catalogue # NBP2-38131PEP) corresponding to the specific epitopes for the LDH-A and LDH-B antibodies (Table 2), respectively, were used for the peptide blocking assays. The respective optical density (O.D) values from the peptide blocking assays were used for experimental blank correction. Data are expressed as experimental blank-corrected O.D_405_ (λ_405_) values from three technical replicates for each of the four biological replicates belonging to each experimental group (*n* = 4).

### 4.6. Quantitative Measurement of Cleaved-Caspase-3, Cleaved-PARP, and Cleaved-DFF45

The levels of *cleaved*-caspase-3 (p19 and p17 fragments), *cleaved*-PARP (p89 fragment), and *cleaved*-DFF45 (p11 fragment) in the experimental lysates were determined, as a surrogate measure of active caspase-3, using a sandwich ELISA immunoassay coupled to a biotin-streptavidin-HRP detection system. Briefly, 20–30 ng of the respective capture antibodies (Table 2) were immobilized in each well of the respective 96-well microplates [98]. The experimental cell lysates (equivalent to 75 μg of protein content) were incubated, with the immobilized respective capture antibodies, overnight at 4 °C. The conditioned cell lysates were discarded, and the 96-well microplates were washed 3× (15 min each) with TBS-T and incubated, with the respective *biotinylated* detection antibodies (Table 2), overnight at 4 °C. The respective 96-well microplates were washed 3× (15 min each) with TBS-T followed by immunodetection with the streptavidin-HRP (streptavidin, HRP conjugate, Sigma Aldrich/Merck Life Science, Darmstadt, Germany, Catalogue # 18-152) using the HRP-substrate Amplex Red (10-acetyl-3,7-dihydroxyphenoxazine) (Thermo Fisher Scientific, Oslo, Norway, Catalogue # A22188) as the chromophore for the colorimetric read-out (λ_570_). The antibody and signal specificity were established by performing peptide blocking assays (with antibody blocking peptide, Table 2) and *immuno-competition* assays with the *unlabeled* detection antibodies to exhibit assay specificity and serve as an experimental blank. The respective absorbances from the peptide blocking assay or *immuno-competition* assays were used for experimental blank correction. Data are expressed as experimental blank-corrected O.D_570_ (λ_570_) values from three technical replicates for each of the four biological replicates belonging to each experimental group (*n* = 4).

### 4.7. Caspase-3 Activity Assay

The enzymatic activity of caspase-3 was measured spectrophotometrically using the specific caspase-3 substrate Ac-DEVD-p-NA (N-Acetyl-Ac-Asp-Glu-Val-Asp-p-nitroanilide) (Sigma Aldrich, Oslo, Norway, Catalogue # 235400-5MG). Caspase-3 activity was measured as a surrogate of the abundance of the released chromophore engendered by the proteolysis of Ac-DEVD-p-NA by caspase-3 [99]. Briefly, AC-16 cells that were terminally sub-cultured and plated in 100 mm cell-culture plates to the desired confluence (4 × 10^6^ cells per plate) and subjected to the respective transfection and experimental interventions were trypsinized and pelleted by centrifugation (1000× *g* for 5 min). The pelleted cells were resuspended in the cell lysis buffer (50 mM HEPES, 5 mM CHAPS, pH 7.4) and incubated on ice for 10 min to lyse the cells, followed by centrifugation at 12,000× *g* for 15 min to pellet the cell debris. The supernatant containing the non-denatured cell lysates (devoid of protease inhibitors) was the dedicated input for caspase-3 activity determination. The protein content of the non-denatured cell lysates was adjusted to a 4 µg/µL concentration and the volume was adjusted to 180 µL with the addition of the assay buffer (20 mM HEPES, 1.62 mM CHAPS, 10 mM Nacl, 2 mM EDTA, pH 7.4). A 200 µg equivalent of protein content (50 µL of the lysate at 4 µg/µL diluted to 180 µL) per well of a 96-well microplate was used as the input from each experimental sample for the caspase-3 activity assay. The input from the respective samples was incubated with the caspase-3 substrate, Ac-DEVD-p-NA (200 µM, 20 µL of 2 mM in a 200 µL assay volume), for 4 h at 37 °C. The caspase-3 activity assays were also performed in the presence of the caspase-3 inhibitor, Ac-DEVD-CHO [99] (Sigma Aldrich/Merck Millipore/Merck Life Science, Darmstadt, Germany, Catalogue # 235420), to exhibit assay specificity and serve as an experimental blank. The absorbance at λ_405_ (405 nm) corresponding to the amount of the chromophore engendered as a commensurate measure of caspase-3 activity was determined using a microplate reader. The respective absorbances from the caspase-3 inhibitor-treated lysates were used for experimental blank correction. Data are expressed as experimental blank-corrected O.D_405_ (λ_405_) values from three technical replicates for each of the four biological replicates belonging to each experimental group (*n* = 4).

### 4.8. Terminal Deoxynucleotidyl Transferase dUTP Nick End Labeling (TUNEL) Assay

The magnitude of DNA fragmentation as a morphological hallmark of *late apoptosis* was determined using an in situ quantitative colorimetric apoptosis detection system “HT TiterTACS™ Apoptosis Detection Kit” from R & D Systems (R & D Systems, Minneapolis, MN, USA, Catalogue # 4822-96-K), following the manufacturer’s guidelines and well-established contemporary protocols [100,101,102,103]. Briefly, AC-16 cells that were terminally sub-cultured and plated in 96-well microplates to the desired confluence (5 × 10^4^ cells/well) and subjected to the respective transfection and experimental interventions (as enunciated earlier) were fixed in situ and subsequently *3′-hydroxyl nick-end-labeled* with biotin-conjugated dNTPs (deoxynucleotide triphosphates) using the DNA polymerase, Terminal deoxynucleotidyl Transferase (TdT) [65]. The *3′-hydroxyl nick-end*-incorporated biotin-conjugated dNTP were detected with streptavidin-conjugated HRP using the HRP-substrate TACS-Sapphire as a chromophore for the colorimetric read-out (λ_450_). The assays performed with *unlabeled* experimental samples, devoid of TdT in the labeling mix, were used to establish signal specificity and serve as experimental blank correction. Data are expressed as experimental blank-corrected O.D_450_ (λ_450_) values from three technical replicates for each of the four biological replicates belonging to each experimental group (*n* = 4).

### 4.9. DFF40 (DFFβ, CAD) Endonuclease Activity Assay

The endonuclease activity of DFF40 (DFFβ, CAD) was determined as a direct molecular correlate of DNA fragmentation. The magnitude of *nicks* in the 3′-hydroxyl termini of double stranded DNA was considered to be a surrogate measure of the endonuclease activity DFF40 (DFFβ) [104,105]. Briefly, nuclear lysates (50 μg of protein equivalent) from experimental AC-16 cells were incubated in reaction buffer (50 mM NaCl, 10 mM HEPES, 2 mM MgCl_2_, 5 EGTA, 1 DTT, 40 mM β–glycerophosphate, 2 mM ATP, 10 mM creatine phosphate, 50 μg/mL creatine kinase, pH 7.0) with 4 μg of genomic DNA, isolated from naïve AC-16 cells, overnight at 32 °C [105]. The endonuclease activity of DFF40 (DFFβ, CAD), present in the nuclear extracts, was quenched by heat-inactivation at 60 °C for 1 h. Subsequently, the conditioned genomic DNA was extracted using a “QIAamp MinElute ccfDNA Midi Kit” (Qiagen Norge, Oslo, Norway, Catalogue # 55284) and resuspended in TE buffer (10 mM Tris, 1 mM EDTA, pH 7.4). The conditioned genomic DNA (3 μg) was biotinylated followed by affinity-capture immobilization in streptavidin-coated 96-well microplates. The conditioned genomic DNA was *3′-hydroxyl nick-end-labeled* with Digoxigenin-conjugated dUTP (deoxyuridine triphosphate) (Digoxigenin-11-dUTP, alkali-stable, Merck Life Science, Darmstadt, Germany, Catalogue # 11558706910) using the DNA polymerase, Terminal deoxynucleotidyl Transferase (TdT) (Thermo Fisher Scientific, Oslo, Norway, Catalogue # EP0161) [65]. The *3′-hydroxyl nick-end*-incorporated Digoxigenin-conjugated dUTP were immunodetected with the AP (alkaline phosphatase)-conjugated digoxigenin antibody (Digoxigenin AP-conjugated Antibody, R&D Systems, Minneapolis, MN, USA, Catalogue # APM7520) using the AP-substrate PNPP (Thermo Fisher Scientific, Oslo, Norway, Catalogue # 37621) as the chromophore for the colorimetric read-out (λ_405_). The assays performed with respective *unlabeled* dUTP-incorporated samples were used to establish signal specificity and serve as experimental blank correction. Data are expressed as experimental blank-corrected O.D_405_ (λ_405_) values from three technical replicates for each of the four biological replicates belonging to each experimental group (*n* = 4).

### 4.10. Cytochrome C Release Assay

The translocation of *Cytochrome C* from the inter-mitochondrial space into the cytosol was determined using a mitochondrial fractionation kit “Cytochrome C Release Apoptosis Assay Kit” from Sigma Aldrich/Merck Millipore (Merck Life Science, Darmstadt, Germany, Catalogue # QIA87), following the manufacturer’s protocol and guidelines. Briefly, AC-16 cells that were terminally sub-cultured, plated in 100 mm cell-culture plates to the desired confluence (4 × 10^6^ cells/plate), and subjected to the respective transfection and experimental interventions (as enunciated earlier) were trypsinized and pelleted by centrifugation (1000× *g* for 5 min). The pelleted cells were resuspended in cytosolic extraction buffer (supplied with the kit) containing DTT (Dithiothreitol) as well as protease inhibitors (Halt™ Protease Inhibitor Cocktail 100×, Thermo Fisher Scientific, Oslo, Norway, Catalogue # 78429) and incubated on ice for 10 min to lyse the cells, followed by centrifugation at 12,000× *g* for 30 min to pellet the cell debris. The resultant supernatant was collected as the cytosolic fraction, while the resultant pellet was resuspended in mitochondrial extraction buffer (supplied with the kit), containing DTT as well as protease inhibitors, to generate the mitochondrial fraction. The respective fractions were processed for Western blot analysis to determine the integrity of the respective fractions. The abundance of *Cytochrome C* in the respective fractions was measured using a sandwich ELISA immunoassay approach. Briefly, 20 ng of the *Cytochrome C* capture antibody (Table 2) was immobilized in each well of a 96-well microplate [98]. The respective cytosolic fractions (equivalent to 50 μg of protein content) and mitochondrial fractions (equivalent to 20 μg of protein content) were incubated, with the immobilized *Cytochrome C* capture antibody, overnight at 4 °C. The conditioned cytosolic fractions and mitochondrial fractions were discarded, and the 96-well microplate wells were washed 3× (15 min each) with TBS-T and incubated with the *Cytochrome C* detection antibody (Table 2) overnight at 4 °C. The 96-well microplate wells were washed 3× (15 min each) with TBS-T, followed by immunodetection with the HRP-conjugated secondary antibody using the HRP-substrate OPD (o-phenylenediamine dihydrochloride) (Thermo Fisher Scientific, Oslo, Norway, Catalogue # 34005) as a chromophore for the colorimetric read-out (λ_450_). The antibody and signal specificity were established by performing peptide blocking assay in the entire gamut of experimental lysates. The *Cytochrome C antibody blocking peptide* (Cell Signaling Technology, Danvers, MA, USA, Catalogue # 1033) corresponding to the specific epitope for the *Cytochrome C* antibody (Table 2) was used for the peptide blocking assay. The respective absorbances from the peptide blocking assay were used for experimental blank correction. Data are expressed as experimental blank-corrected O.D_450_ (λ_450_) values from three technical replicates for each of the four biological replicates belonging to each experimental group (*n* = 4).

### 4.11. Cellular Fractionation to Segregate the Cytosolic and Mitochondrial Compartments

The cytosolic and mitochondrial fractions were isolated using a “Mitochondria/Cytosol Fractionation Kit” from Abcam (Abcam, Cambridge, UK, Catalogue # ab65320), following the manufacturer’s protocol and guidelines. Briefly, AC-16 cells that were terminally sub-cultured, plated in 150 mm cell-culture plates to the desired confluence (2 × 10^7^ cells/plate), and subjected to the respective transfection and experimental interventions (as enunciated earlier) were trypsinized and pelleted by centrifugation (1000× *g* for 5 min at 4 °C). The pelleted cells were resuspended in 1× Cytosol Extraction Buffer Mix containing DTT (dithiothreitol) and protease inhibitors (supplied with the kit), followed by incubation on ice for 10 min. The cells were homogenized in a Dounce tissue grinder on ice with 30–50 passes of the grinder. The resulting cell homogenate (lysate) was transferred to a 1.5-mL microcentrifuge tube and centrifuged at 700× *g* for 10 min at 4 °C. The resulting supernatant (pellet discarded) was transferred to a fresh 1.5 mL microcentrifuge tube and centrifuged at 10,000× *g* for 30 min at 4 °C. The resulting supernatant constituted the cytosolic fraction while the resultant pellet constituted the mitochondrial fraction. The pellet constituting the mitochondrial fraction was either resuspended in 100 μL 1× PBS (pH 7.4) to yield intact mitochondria or, as desired, mitochondrial protein lysate resuspended in 100 μL of the Mitochondrial Extraction Buffer Mix containing DTT and protease inhibitors (provided with the kit) to yield the total mitochondrial lysate fraction.

### 4.12. Isolation of Heavy Mitochondrial Fractions for ELISA Immunoassays

Heavy mitochondrial fractions were prepared using a “Mitochondria Isolation Kit” from Sigma Aldrich/Merck Millipore (Merck Life Science, Darmstadt, Germany, Catalogue # MITOISO2), following the manufacturer’s protocol and guidelines. Briefly, AC-16 cells that were terminally sub-cultured, plated in 150 mm cell-culture plates to the desired confluence (2 × 10^7^ cells/plate), and subjected to the respective transfection and experimental interventions (as enunciated earlier) were trypsinized and pelleted by centrifugation (600× *g* for 5 min at 4 °C). The pelleted cells were resuspended in 1.2 mL lysis buffer (supplied with the kit) containing protease inhibitors (supplied with the kit) and incubated on ice for 5 min to lyse the cells, followed by the addition of the 2× volume of extraction buffer (supplied with the kit) and centrifugation at 1000× *g* for 10 min at 4 °C. The resultant supernatant was further centrifuged at 3500× *g* for 10 min at 4 °C to generate a heavy mitochondrial fraction. For quantitative ELISA immunoassays, the pelleted heavy mitochondrial fraction was resuspended in 200 μL of CelLytic^TM^ M Cell Lysis Reagent (supplied with the kit) containing protease inhibitors (1:100 (*v*/*v*)). The quantitative ELISA immunoassays used to determine the abundance of BCL2 family of proteins were performed as enunciated and described in Section 4.12.

### 4.13. Quantitative Measurement of Members of the BCL2 Family of Proteins by Sandwich ELISA

The levels of BAX, BAK, BCL2, BCL-X_L_, and MCL1 in the mitochondrial and cytosolic fractions, as well as whole-cell lysates, were determined using sandwich ELISA immunoassay. Briefly, 10–30 ng of the respective capture antibodies (Table 2) were immobilized in each well of the respective 96-well microplates [98]. The respective mitochondrial fractions (equivalent to 20 μg of protein content) and cytosolic fractions (equivalent to 40 μg of protein content) were incubated with the respective immobilized capture antibodies overnight at 4 °C. The conditioned mitochondrial fractions and cytosolic fractions were discarded, and the respective 96-well microplate wells were washed 3× (15 min each) with TBS-T and incubated with the respective detection antibodies (Table 2) overnight at 4 °C. The 96-well microplate wells were washed 3× (15 min each) with TBS-T followed by immunodetection with the HRP-conjugated secondary antibody, using the HRP-substrate OPD (Thermo Fisher Scientific, Oslo, Norway, Catalogue # 34005) as a chromophore for the colorimetric read-out (λ_450_). The antibody signal specificity was established by performing peptide blocking assays in the entire gamut of mitochondrial fractions and cytosolic fractions from experimental cells. The antibody blocking peptides corresponding to the specific epitopes for the respective detection antibodies used are enumerated in Table 2. The respective absorbances from the peptide blocking assays were used for the experimental blank correction. Data are expressed as experimental blank-corrected O.D_450_ (λ_450_) values from three technical replicates for each of the four biological replicates belonging to each experimental group (*n* = 4).

### 4.14. Isolation and Preparation of Alkali-Resistant OMM (Outer Mitochondrial Membrane): Inserted and Alkali-Soluble OMM-Tethered Protein Fractions for the Quantitative Measurement of OMM-Inserted and OMM-Tethered BAX and BAK

To determine the abundance of oligomeric *OMM-inserted* BAX and BAK, the pelleted heavy mitochondrial fraction (from the isolated heavy mitochondrial fractions described in Section 4.12) was resuspended and incubated in 900 μL of 100 mM sodium carbonate (pH 11.5) at 4 °C for 45 min, followed by ultracentrifugation at 144,000× *g* at 4 °C for 45 min [77,78,79]. The pellet constituted the alkali-resistant *OMM-inserted* protein fraction containing the *OMM-inserted* BAX and BAK oligomers while the supernatant constituted the alkali-soluble *OMM-tethered* protein fraction containing the *OMM-tethered* BAX and BAK monomers. The supernatants, constituting the *OMM-tethered* protein fraction from respective samples, were neutralized with glacial acetic acid and the alkali-soluble proteins were TCA (trichloroacetic acid)-precipitated [106] followed by centrifugation at 15,000× *g* at 4 °C for 60 min. The pellets emanating from both the alkali-resistant *OMM-inserted* protein fraction (containing the *OMM-inserted* BAX and BAK oligomers) and the alkali-soluble *OMM-tethered* protein fraction (containing the *OMM-tethered* BAX and BAK monomers) were resuspended in denaturing RIPA lysis buffer (Tris 50 mM, sodium chloride (Nacl) 150 mM, Sodium Deoxycholate 0.5% *w*/*v*, sodium dodecyl sulphate (SDS) 0.1% *w*/*v*, Triton-X 1% *v*/*v*, pH 7.4) supplemented with protease and phosphatase inhibitors (Halt^TM^ Protease and Phosphatase Inhibitor Cocktail 100×, Thermo Fisher Scientific, Oslo, Norway, Catalogue # 78446). The fractions were further processed for ELISA immunoassays to determine the relative abundance of BAX and BAK in the respective fractions as well as validation of the purity and integrity of the respective fractions. The validation of the purity and integrity of the respective fractions was performed by analyzing the respective fractions for the presence and abundance of the following: TOM40, an *OMM-inserted* protein; TIM22, an *IMM* (inner mitochondrial membrane)*-inserted* protein; HK2, an *OMM-tethered* protein, and SDHA, an *IMM-tethered* protein. The quantitative ELISA immunoassays that determined the relative abundance of BAX and BAK in the respective fractions was performed as described in Section 4.13.

### 4.15. Statistical Analysis

The significance of differences among the samples was determined by *one-way analysis of variance* (one-way ANOVA) followed by Tukey’s *post hoc* test. Statistical analysis was performed with GraphPad Prism 8 (GraphPad Software, San Diego, CA, USA). Quantitative data for all the assays are presented as mean values ± S.D (standard deviation).

## 5. Conclusions

Our unprecedented findings that miR-210-induced inhibition of GSK3β kinase activity underpins the miR-210-elicited alleviation of apoptotic cell death, further broadens the gamut of the spectrum of the biological effects, and widens the repertoire of the identified molecular targets of miR-210 in the adaptive cellular response to hypoxia stress. Our unprecedented findings unveil the presence of a unique miR-210-GSK3β signaling cross-talk that is intricately involved in the regulation of hypoxia-evoked intrinsic apoptosis cascade. Our study is the first to expound on and delineate the downstream molecular targets of this miR-210-GSK3β signaling cross-talk and cogently demonstrate that the miR-210 mitigates the hypoxia-induced BAX translocation to the mitochondria and the ensuing *OMM*-insertion of BAX and BAK oligomers, as well as the ensuing release of mitochondrial *Cytochrome C* into the cytosol. Our study provides an invaluable insight into the molecular underpinnings of miR-210 modulation of hypoxia-induced intrinsic apoptosis cascade and ushers in a new frontier in our comprehension of novel molecular targets in the quest to design therapeutic approaches to combat the ravages of IHD.

## Figures and Tables

**Figure 1 ijms-23-09375-f001:**
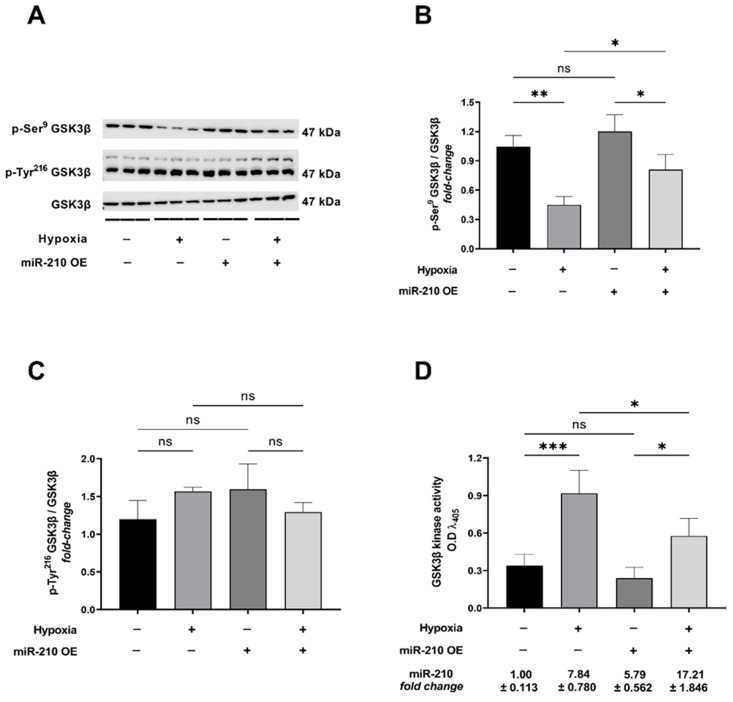
miR-210 attenuates the hypoxia-induced increase in GSK3β kinase activity. (**A**–**C**) Representative Western blots (**A**) and quantitative densitometric analysis determining the *inhibitory* phosphorylation of GSK3β at the Ser^9^ residue (**B**) concomitant with the *activating* phosphorylation of GSK3β at the Tyr^216^ residue (**C**). (**D**) Quantitative GSK3β kinase activity assay in native non-denatured lysates. miR-210 expression levels in the corresponding respective cell lysates were determined by the miR-210 hybridization immunoassay (as described in Section 4.3). Data from the Western blot and densitometric analysis are expressed as mean *fold-change* ± S.D from three biological replicates belonging to each experimental group (*n* = 3). Data from the GSK3β activity assay are expressed as experimental blank-corrected absorbances (O.D) measured at λ_405_ (405 nm). Data from the GSK3β activity assay are expressed as mean ± S.D from three technical replicates for each of the four biological replicates belonging to each experimental group (*n* = 4). miR-210 expression levels are depicted as *fold-change* ± S.D. * *p* ≤ 0.05; ** *p* ≤ 0.01; *** *p* ≤ 0.001; ns: not significant (*p* > 0.05). OE: miR-210 overexpression; O.D: optical density; S.D: standard deviation.

**Figure 2 ijms-23-09375-f002:**
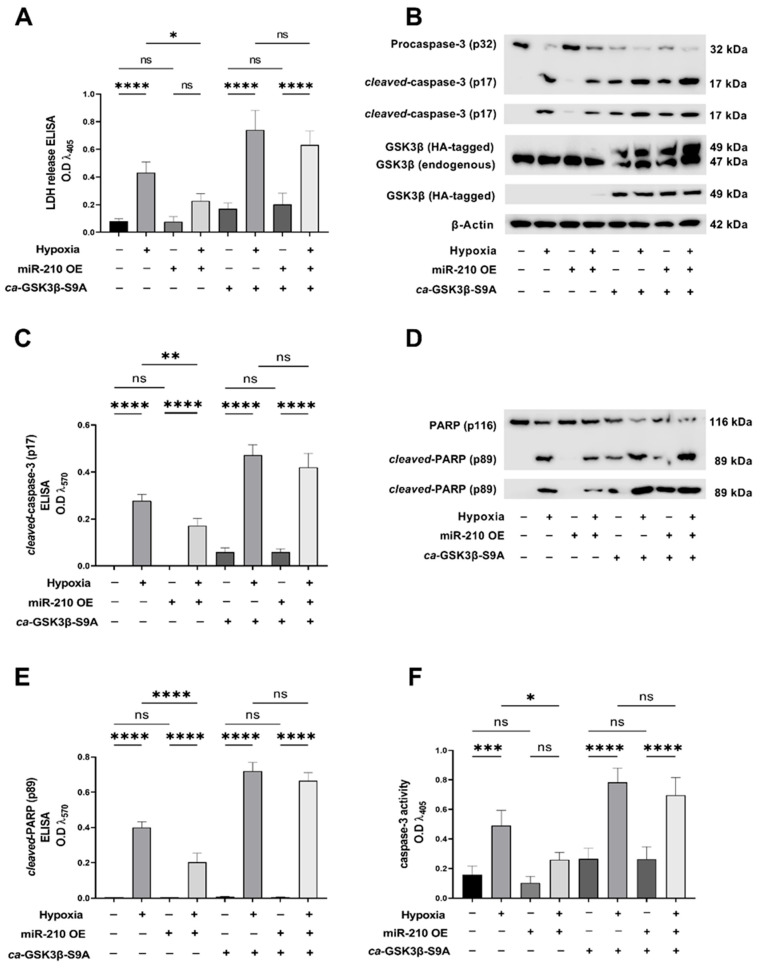
miR-210 mitigates the hypoxia-induced caspase-3 activation-mediated cell death through the inhibition of GSK3β kinase activity. (**A**) Quantitative ELISA determining the LDH release in the conditioned media as a surrogate marker of cell death. Data from the LDH ELISA are expressed as experimental blank-corrected absorbances (O.D) measured at λ_450_ (450 nm). (**B**) Qualitative Western blot depicting the processing of the inert procaspase-3 (p32) into the active cleaved-caspase-3 (p17 fragment) as a surrogate measure of caspase-3 activation status (**C**) Quantitative ELISA determining the levels of the active cleaved-caspase-3 (p17 fragment), expressed as experimental blank-corrected absorbances (O.D) measured at λ_570_ (570 nm). (**D**) Qualitative Western blot depicting the processing of the caspase-3 substrate PARP (p116) into the *cleaved*-PARP (p89 fragment) as a surrogate measure of caspase-3 activation status. (**E**) Quantitative ELISA determining the levels of the *cleaved*-PARP (p89 fragment), expressed as experimental blank-corrected absorbances (O.D) measured at λ_570_ (570 nm). (**F**) Quantitative caspase-3 activity assay in native non-denatured lysates, expressed as experimental blank-corrected absorbances (O.D) measured at λ_405_ (405 nm). miR-210 expression levels in the respective cell lysates were determined by the miR-210 hybridization immunoassay (as described in Section 4.3) and are reported in Appendix A. The validation of the ectopic expression of the HA-tagged *ca*-GSK3β-S9A mutant in the pertinent experimental groups was performed by Western blot analysis (**B**) as well as ELISA immunoassay and is reported in Appendix A. GSK3β kinase activity (as described in Section 4.4) was measured in all experimental groups to corroborate and validate the translative effects of the ectopic expression of the *ca*-GSK3β-S9A mutant (reported in Appendix A). All data are expressed as *mean ± S.D* from three technical replicates for each of the four biological replicates belonging to each experimental group (*n* = 4). * *p* ≤ 0.05; ** *p* ≤ 0.01; *** *p* ≤ 0.001; **** *p* ≤ 0.0001; ns: not significant (*p* > 0.05). OE: miR-210 overexpression; O.D: optical density; S.D: standard deviation.

**Figure 3 ijms-23-09375-f003:**
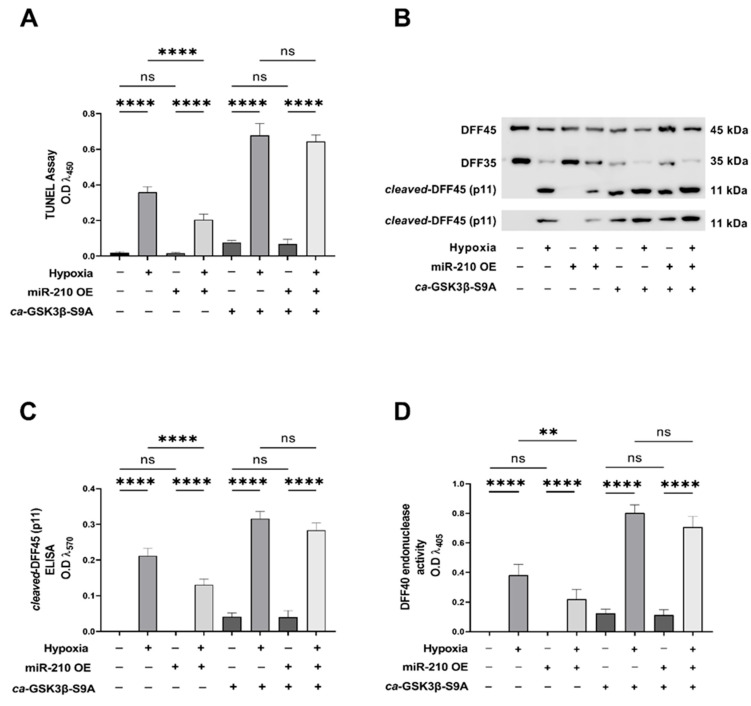
miR-210 attenuates the hypoxia-induced increase in DNA fragmentation through the inhibition of GSK3β kinase activity. (**A**) Quantitative TUNEL assay determining apoptotic DNA fragmentation, expressed as experimental blank-corrected absorbances (O.D) measured at λ_450_ (450 nm). (**B**) Qualitative Western blot depicting the processing of the DFF45 into *cleaved*-DFF45 (p11 fragment) as a surrogate measure of caspase-3-induced DFF40 endonuclease activity. (**C**) Quantitative ELISA determining the levels of *cleaved*-DFF45 (p11 fragment), expressed as experimental blank-corrected absorbance (O.D) measured at λ_570_ (570 nm). (**D**) Quantitative DFF40 endonuclease activity assay, expressed as experimental blank-corrected absorbances (O.D) measured at λ_405_ (405 nm). miR-210 expression levels in the respective cell lysates were determined by the miR-210 hybridization immunoassay (as described in Section 4.3) and are reported in Appendix A. The validation of the ectopic expression of the HA-tagged *ca*-GSK3β-S9A mutant in the pertinent experimental groups was performed by ELISA immunoassay and is reported in Appendix A). GSK3β kinase activity (as described in Section 4.4) was measured in all experimental groups to corroborate and validate the translative effects of the ectopic expression of the *ca*-GSK3β-S9A mutant (reported in Appendix A). All data are expressed as mean ± S.D from three technical replicates for each of the four biological replicates belonging to each experimental group (*n* = 4). ** *p* ≤ 0.01; **** *p* ≤ 0.0001; ns: not significant (*p* > 0.05). OE: miR-210 overexpression; O.D: optical density; S.D: standard deviation.

**Figure 4 ijms-23-09375-f004:**
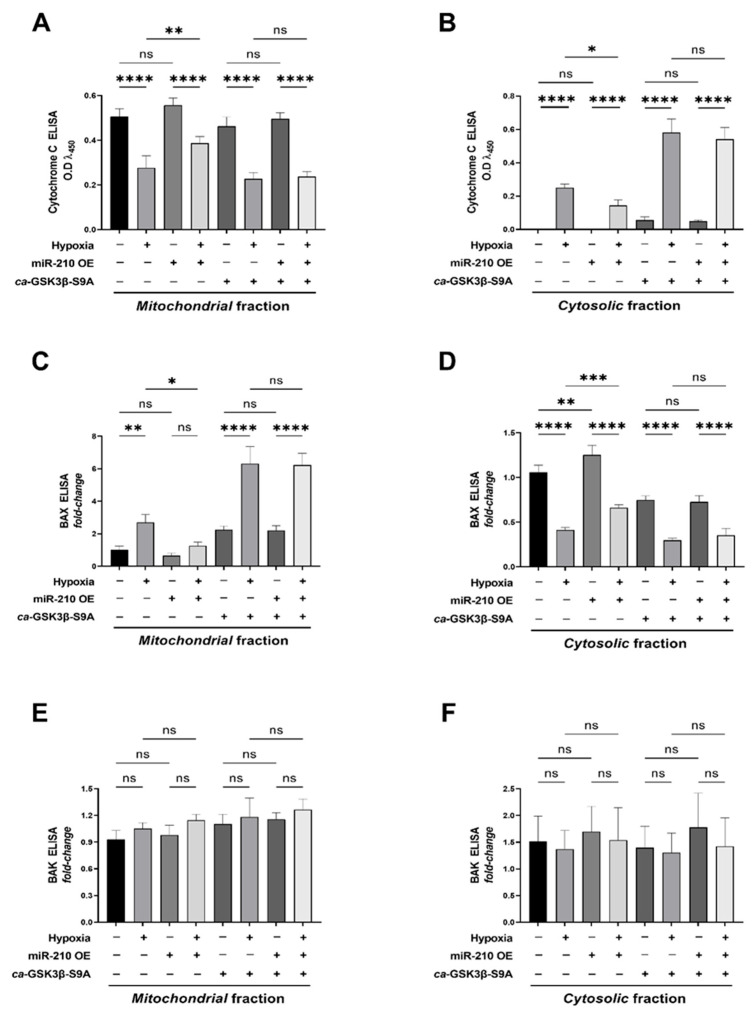
miR-210 attenuates the hypoxia-driven *intrinsic apoptosis pathway* through the inhibition of GSK3β kinase activity. (**A**,**B**) Quantitative ELISA immunoassays determining the *Cytochrome C* abundance in the mitochondrial fractions (A) and cytosolic fractions (**B**), expressed as experimental blank-corrected absorbances (O.D) measured at λ_450_ (450 nm). (**C**–**F**) Quantitative ELISA immunoassays determining the abundance of BAX (**C**,**D**) and BAK (**E**,**F**) in the mitochondrial fractions (**C**,**E**) and cytosolic fractions (**D**,**F**), expressed as experimental blank-corrected absorbances (O.D) measured at λ_450_ (450 nm) normalized to *fold-change* values. Quantitative ELISA immunoassays determining the abundance of BAX and BAK in whole-cell fractions are reported in Appendix A. The validity of the integrity of the respective subcellular compartments is reported in Appendix A. miR-210 expression levels in the respective cell lysates were determined by the miR-210 hybridization immunoassay (as described in Section 4.3) and are reported in Appendix A. The validation of the ectopic expression of the HA-tagged *ca*-GSK3β-S9A mutant in the pertinent experimental groups was performed by ELISA immunoassay and is reported in Appendix A. GSK3β kinase activity (as described in Section 4.4) was measured in all experimental groups to corroborate and validate the translative effects of the ectopic expression of the *ca*-GSK3β-S9A mutant (reported in Appendix A). All data are expressed as *mean ± S.D* from three technical replicates for each of the four biological replicates belonging to each experimental group (*n* = 4). * *p* ≤ 0.05; ** *p* ≤ 0.01; *** *p* ≤ 0.001; **** *p* ≤ 0.0001; ns: not significant (*p* > 0.05). OE: miR-210 overexpression; O.D: optical density; S.D: standard deviation.

**Figure 5 ijms-23-09375-f005:**
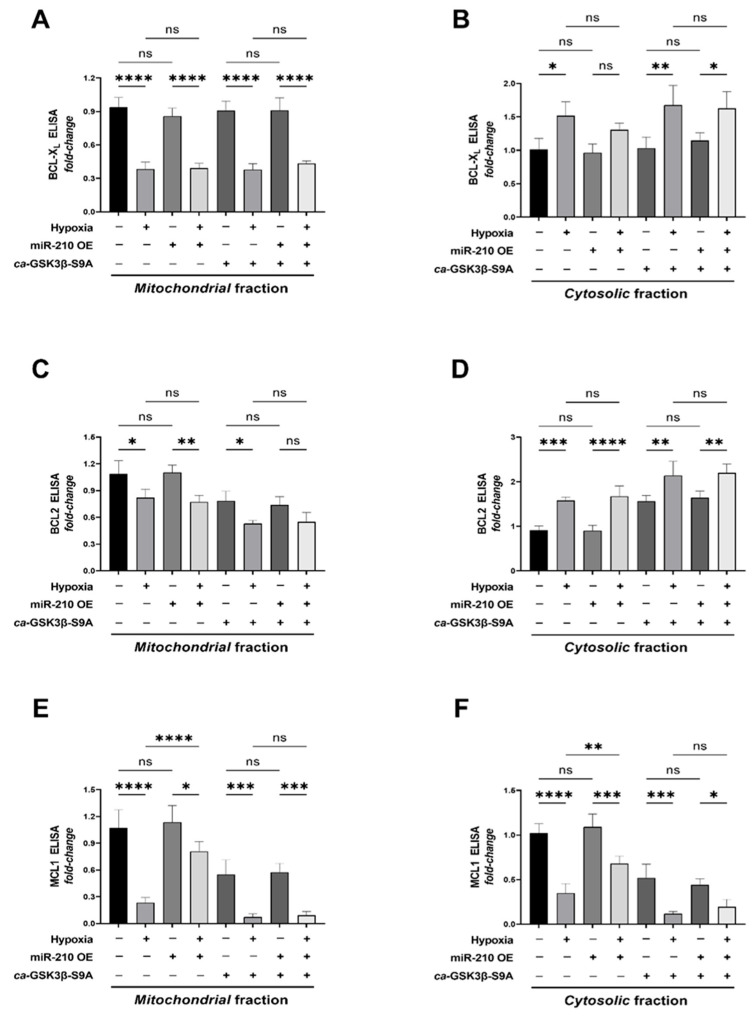
miR-210 mitigates the hypoxia-driven decrease in MCL1 levels in the mitochondria through the inhibition of GSK3β kinase activity. (**A**–**F**) Quantitative ELISA immunoassays determining the abundance of the BCL2 family of antiapoptotic proteins, BCL-X_L_ (**A**,**B**), BCL2 (**C**,**D**), and MCL1 (**E**,**F**), in the mitochondrial fractions (**A**,**C**,**E**) and the cytosolic fractions (**B**,**D**,**F**). Data are expressed as experimental blank-corrected absorbances (O.D) measured at λ_450_ (450 nm) normalized to *fold-change* values. Quantitative ELISA immunoassays determining the abundance of BCL-X_L_, BCL2, and MCL1 in the *whole-cell fractions* are reported in Appendix A. miR-210 expression levels in the respective cell lysates were determined by the miR-210 hybridization immunoassay (as described in Section 4.3) and are reported in Appendix A. The validation of the ectopic expression of the HA-tagged *ca*-GSK3β-S9A mutant in the pertinent experimental groups was performed by ELISA immunoassay and is reported in Appendix A. GSK3β kinase activity (as described in Section 4.4) was measured in all experimental groups to corroborate and validate the translative effects of the ectopic expression of the *ca*-GSK3β-S9A mutant (reported in Appendix A). All data are expressed as *mean ± S.D fold-change* values from three technical replicates for each of the four biological replicates belonging to each experimental group (*n* = 4). * *p* ≤ 0.05; ** *p* ≤ 0.01; *** *p* ≤ 0.001; **** *p* ≤ 0.0001; ns: not significant (*p* > 0.05). OE: miR-210 overexpression; O.D: optical density; S.D: standard deviation.

**Figure 6 ijms-23-09375-f006:**
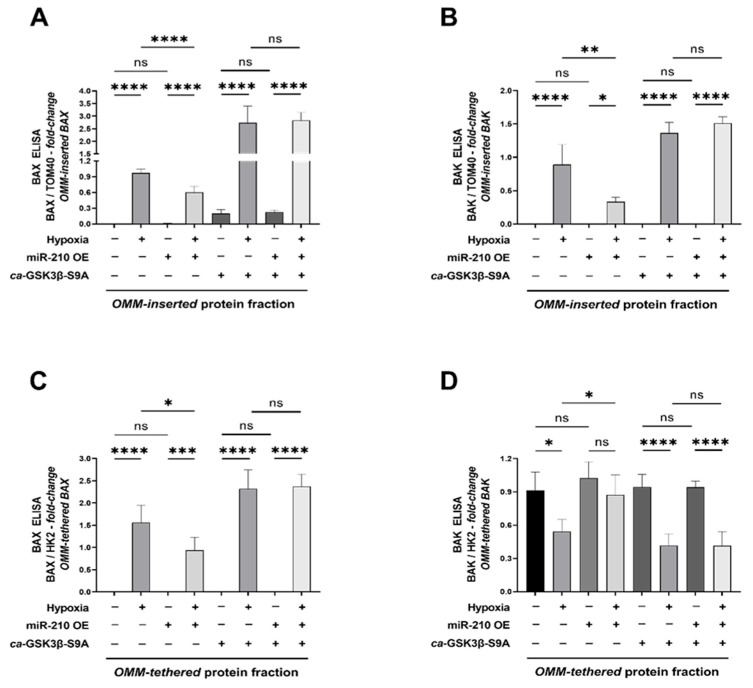
miR-210 mitigates the hypoxia-induced oligomerization and insertion of BAX and BAK into the *outer mitochondrial membrane* (*OMM*). Isolated mitochondria were subjected to a 0.1 M sodium carbonate (Na_2_CO_3_) treatment to produce the alkali-resistant *OMM-inserted* (*OMM*-embedded) protein fraction and the alkali-soluble *OMM-tethered* (*OMM*-anchored) protein fraction. (**A**–**D**) ELISA immunoassays determining the abundance of BAX (**A**,**C**) and BAK (**B**,**D**) performed on the *denatured* lysates from the respective *OMM-inserted* protein fraction (**A**,**B**) and *OMM-tethered* protein fraction (**C**,**D**). Experimental blank-corrected absorbances (O.D) measured at λ_450_ (450 nm) were first expressed as *fold-change* and subsequently normalized to either TOM40 expression levels (for the *OMM-inserted* protein fraction) or HK2 expression levels (*OMM-tethered* protein fraction). The expression levels of TOM40 and HK2 in the respective fractions are reported in Appendix A. miR-210 expression levels in the respective whole-cell lysates were determined by the miR-210 hybridization immunoassay (as described in Section 4.3) and are reported in Appendix A. The validation of the ectopic expression of the HA-tagged *ca*-GSK3β-S9A mutant in the pertinent experimental groups was performed by ELISA immunoassay and is reported in Appendix A. GSK3β kinase activity (as described in Section 4.4) was measured in all experimental groups to corroborate and validate the translative effects of the ectopic expression of the *ca*-GSK3β-S9A mutant (reported in Appendix A). Data are represented as double-normalized ratiometric values (BAX/TOM40 and BAX/HK2 as well as BAK/TOM40 and BAK/HK2), expressed as *mean ± S.D fold-change*, from three technical replicates for each of the four biological replicates belonging to each experimental group (*n* = 4). * *p* ≤ 0.05; ** *p* ≤ 0.01; *** *p* ≤ 0.001; **** *p* ≤ 0.0001; ns: not significant (*p* > 0.05). OE: miR-210 overexpression; O.D: optical density; S.D: standard deviation; *OMM*: outer mitochondrial membrane.

**Figure 7 ijms-23-09375-f007:**
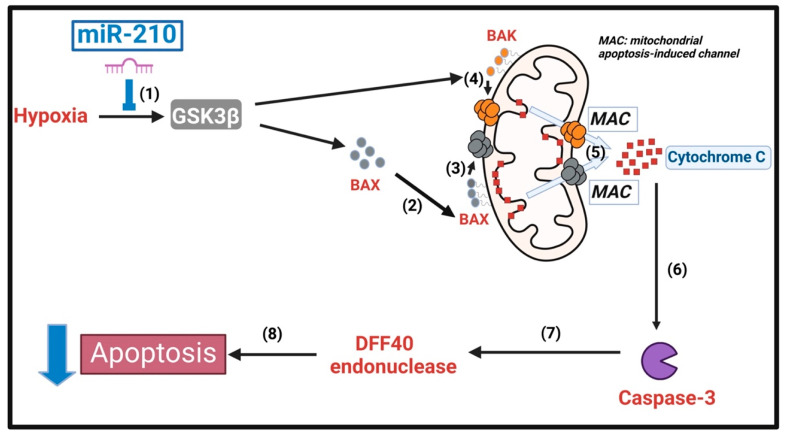
An illustrated schematic depicting the inhibition of GSK3β kinase activity as the molecular pivot that underlies the miR-210-elicited mitigation of hypoxia-induced intrinsic apoptosis cascade. miR-210 attenuates the hypoxia-induced GSK3β kinase activity (1) that decreases the hypoxia-induced increase in the abundance of the OMM (outer mitochondrial membrane)-tethered pool of BAX (2). Subsequently, the miR-210-elicited inhibition of the hypoxia-induced increase in GSK3β kinase activity results in a significant decrease in the hypoxia-induced abundance of the OMM-inserted pool of BAX (3) and BAK (4) that constitute the MAC (mitochondrial apoptosis-induced channel). The miR-210-elicited inhibition of the hypoxia-induced increase in GSK3β kinase activity translates into a decrease in the hypoxia-induced abundance of MAC formation that culminates in a commensurate mitigation of the hypoxia-induced increase in Cytochrome C release from the mitochondria into the cytosol (5). The miR-210-evoked GSK3β inhibition-mediated reduction in the hypoxia-induced increase in Cytochrome C release into the cytosol translates into a decrease in the hypoxia-induced caspase-3 activity (6) and the ensuing DFF40 endonuclease activity (7) that culminates in the attenuation of hypoxia-induced apoptotic cell death (8). This illustration was created using BioRender.com (https://app.biorender.com/illustrations/627d566667f7b4db938c909f, accessed on 8 August 2022).

**Table 1 ijms-23-09375-t001:** Experimental paradigm and experimental groups.

	Control Empty Vector (pEZX-MR04-Scrambled)	miR-210 Overexpression (OE) Vector (pEZX-MR04-miR-210)	Control Empty Vector (pcDNA3)	HA-Tagged GSK3 Beta S9A (pcDNA3-HA-GSK3β S9A)
Normoxia, 18 h	*n* = 4	*n* = 4	*n* = 4	*n* = 4
Hypoxia, 18 h	*n* = 4	*n* = 4	*n* = 4	*n* = 4

*n* = 4: Four biological replicates.

**Table 2 ijms-23-09375-t002:** List of antibodies and antibody-blocking peptides used in the study.

Antibody	Application	Amount	Host	Manufacturer	Catalogue #	Resource Identifier ID (RRID)
β-Actin	WB 1:5000	1 µg	Mouse	Santa Cruz Biotechnology	sc-47778	AB_2714189
β-Actin	ELISA *capture*	20 ng/well	Mouse	Santa Cruz Biotechnology	sc-47778	AB_2714189
β-Actin	ELISA *detection*	20 ng/well	Rabbit	Cell Signaling Technology	4970	AB_2223172
β-Actin antibody blocking peptide	ELISA *detection*	N/A	N/A	Cell Signaling Technology	1025	N/A
BAK	ELISA *capture*	20 ng/well	Mouse	Thermo Fisher Scientific	MA5-36225	AB_2884059
BAK	ELISA *detection*	20 ng/well	Rabbit	Novus Biologicals	NBP1-77152	AB_11014847
BAK antibody blocking peptide	ELISA *detection*	N/A	N/A	Novus Biologicals	NBP1-77152PEP	N/A
BAX	ELISA *capture*	20 ng/well	Mouse	Thermo Fisher Scientific	33-6600	AB_2533133
BAX	ELISA *detection*	20 ng/well	Rabbit	Novus Biologicals	NBP1-88682	AB_11014342
BAX antibody blocking peptide	ELISA *detection*	N/A	N/A	Novus Biologicals	NBP1-88682PEP	N/A
BCL2	ELISA *capture*	20 ng/well	Mouse	Thermo Fisher Scientific	BMS1028	AB_10597451
BCL2	ELISA *detection*	20 ng/well	Rabbit	Thermo Fisher Scientific	PA5-20068	AB_11152761
BCL2 antibody blocking peptide	ELISA *detection*	N/A	N/A	Thermo Fisher Scientific	PEP-0187	N/A
Caspase-3	WB 1:1000	5 µg	Rabbit	Cell Signaling Technology	14220	AB_2798429
*cleaved*-caspase-3 (Asp^175^)	ELISA *capture*	20 ng/well	Rabbit	Cell Signaling Technology	9579	AB_10897512
*cleaved*-caspase-3 (Asp^175^) (biotinylated)	ELISA *detection*	20 ng/well	Rabbit	Cell Signaling Technology	9654	AB_10694088
*cleaved*-caspase-3 (Asp^175^) antibody blocking peptide	ELISA *detection*	N/A	N/A	Cell Signaling Technology	1050	N/A
COX4	WB 1:1000	5 µg	Rabbit	Cell Signaling Technology	4844	AB_2085427
COX4	ELISA *capture*	20 ng/well	Mouse	Thermo Fisher Scientific	MA5-15686	AB_10977841
COX4	ELISA *detection*	20 ng/well	Rabbit	Cell Signaling Technology	4844	AB_2085427
COX4 antibody blocking peptide	ELISA *detection*	N/A	N/A	Cell Signaling Technology	1034	N/A
DFF45	WB 1:500	5 µg	Rabbit	Cell Signaling Technology	9732	AB_329956
*cleaved*-DFF45 (Asp^224^)	WB 1:500	5 µg	Rabbit	Cell Signaling Technology	9731	AB_329954
*cleaved*-DFF45 (Asp^224^)	ELISA *capture*	20 ng/well	Rabbit	Cell Signaling Technology	9731	AB_329954
*cleaved*-DFF45 (Asp^224^) (biotinylated)	ELISA *detection*	20 ng/well	Rabbit	Cell Signaling Technology	9731	AB_329954
Cytochrome C	ELISA *capture*	20 ng/well	Mouse	Thermo Fisher Scientific	BMS1037	AB_10598651
Cytochrome C	ELISA *detection*	20 ng/well	Rabbit	Cell Signaling Technology	4280	AB_10695410
Goat Anti-Mouse IgG (H + L)-HRP Conjugate	1:5000	1 µg	Goat	Bio-Rad	1706516	AB_11125547
Goat Anti-Mouse IgG-AP Conjugate	1:5000	N/A ^€^	Goat	Bio-Rad	1706520	AB_11125348
Goat Anti-Rabbit IgG (H + L)-HRP Conjugate	1:5000	1 µg	Goat	Bio-Rad	1706515	AB_11125142
Goat Anti-Rabbit IgG-AP Conjugate	1:20,000	N/A ^€^	Goat	Sigma Aldrich/Merck Life Science	A3687	AB_258103
GSK3β	WB 1:1000	5 µg	Rabbit	Cell Signaling Technology	9315	AB_490890
p-Ser^9^ GSK3β	WB 1:1000	5 µg	Rabbit	Cell Signaling Technology	9322	AB_2115196
p-Tyr^279^/Tyr^216^ GSK3α/β	WB 1:1000	5 µg	Rabbit	Thermo Fisher Scientific	PA5-36646	AB_2553634
HA tag	WB 1:1000	5 µg	Mouse	Thermo Fisher Scientific	26183	AB_10978021
HA tag	ELISA *capture*	30 ng/well	Mouse	Thermo Fisher Scientific	26183	AB_10978021
HA tag	ELISA *detection*	30 ng/well	Rabbit	Abcam	ab13834	AB_443010
HA tag antibody blocking peptide	ELISA *detection*	N/A	N/A	Abcam	ab13835	N/A
HK2	ELISA *capture*	30 ng/well	Rabbit	Thermo Fisher Scientific	PA5-97828	AB_2812442
HK2	ELISA *detection*	30 ng/well	Mouse	Thermo Fisher Scientific	MA5-15679	AB_10986812
LDH	ELISA *capture*	30 ng/well	Mouse	Santa Cruz Biotechnology	sc-133123	AB_2134964
LDH-A	ELISA *detection*	30 ng/well	Rabbit	Novus Biologicals	NBP1-48336	AB_10011099
LDH-A antibody blocking peptide	ELISA *detection*	N/A	N/A	Novus Biologicals	NBP1-48336PEP	N/A
LDH-B	ELISA *detection*	30 ng/well	Rabbit	Novus Biologicals	NBP2-38131	N/A
LDH-A antibody blocking peptide	ELISA *detection*	N/A	N/A	Novus Biologicals	NBP2-38131PEP	N/A
MCL1	ELISA *capture*	20 ng/well	Mouse	Thermo Fisher Scientific	MA5-15236	AB_10986161
MCL1	ELISA *detection*	20 ng/well	Rabbit	Thermo Fisher Scientific	PA5-20121	AB_11152825
MCL1 antibody blocking peptide	ELISA *detection*	N/A	N/A	Thermo Fisher Scientific	PEP-0239	N/A
PARP	WB 1:1000	5 µg	Rabbit	Cell Signaling Technology	9542	AB_2160739
*cleaved*-PARP (Asp^214^)	WB 1:1000	5 µg	Mouse	Cell Signaling Technology	9546	AB_2160593
*cleaved*-PARP (Asp^214^)	ELISA *capture*	20 ng/well	Mouse	Cell Signaling Technology	9546	AB_2160593
*cleaved*-PARP (Asp^214^) (biotinylated)	ELISA *detection*	20 ng/well	Rabbit	Cell Signaling Technology	9185	AB_10858875
p-Ser	GSK3β activity	100 ng/well	Mouse	Santa Cruz Biotechnology	sc-81516	AB_1128626
SDHA (SDH2)	ELISA *capture*	30 ng/well	Mouse	Thermo Fisher Scientific	459200	AB_2532231
SDHA (SDH2)	ELISA *detection*	30 ng/well	Rabbit	Cell Signaling Technology	11998	AB_2750900
TOM40	ELISA *capture*	30 ng/well	Mouse	Santa Cruz Biotechnology	sc-365467	AB_10847086
TOM40	ELISA *detection*	30 ng/well	Rabbit	Thermo Fisher Scientific	18409-1-AP	AB_2303725
TIM22	ELISA *capture*	30 ng/well	Mouse	Sigma Aldrich/Merck Life Science	SAB1400520-	AB_1858016
TIM22	ELISA *detection*	30 ng/well	Rabbit	Thermo Fisher Scientific	14927-1-AP	AB_11183050

WB: Western blot. N/A: Not available/not applicable. €: Amount of secondary antibody cannot be determined as the commercial vendor does not provide the antibody concentration.

## Data Availability

All data are included in the manuscript. Appendix A can be requested from the corresponding author.

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
