# Peer review of "GSK3β Inhibition Is the Molecular Pivot That Underlies the Mir-210-Induced Attenuation of Intrinsic Apoptosis Cascade during Hypoxia"

_ijms, 2022, doi:10.3390/ijms23169375_

Round 1

Reviewer 1 Report

In this study, authors demonstrated the salutary protective response of miR-210 in mitigating the hypoxia-induced apoptotic cell death via inhibiting the serine/threonine kinase Glycogen Synthase Kinase 3 beta (GSK3β) using AC-16 cardiomyocytes subjected to hypoxia stress (exposure to hypoxia for 18 hours).

It is well known that miR-210 is a hypoxia inducible miR. Authors also demonstrated that the expression of miR-210 was significantly upregulated in AC-16 cardiomyocytes following exposure to hypoxia (Figure 1D). The level of miR-210 in AC-16 is higher than that in cells directly transfected with miR-210 (7.84 ± 0.780 vs 5.79 ± 0.562). Therefore, it is very important to include the results of cell injury and GSK3β kinase activity in AC-16 cardiomyocytes with downregulated miR-210 or blocked miR-210 upregulation following exposure to hypoxia.

Author Response

In this study, authors demonstrated the salutary protective response of miR-210 in mitigating the hypoxia-induced apoptotic cell death via inhibiting the serine/threonine kinase Glycogen Synthase Kinase 3 beta (GSK3β) using AC-16 cardiomyocytes subjected to hypoxia stress (exposure to hypoxia for 18 hours).                                                                                                                    It is well known that miR-210 is a hypoxia inducible miR. Authors also demonstrated that the expression of miR-210 was significantly upregulated in AC-16 cardiomyocytes following exposure to hypoxia (Figure 1D). The level of miR-210 in AC-16 is higher than that in cells directly transfected with miR-210 (7.84 ± 0.780 vs 5.79 ± 0.562). Therefore, it is very important to include the results of cell injury and GSK3β kinase activity in AC-16 cardiomyocytes with downregulated miR-210 or blocked miR-210 upregulation following exposure to hypoxia.

We thank the reviewer for the critical appraisal of the findings reported in this study / manuscript (ijms-1819501). Indeed, the reviewer correctly highlights that the expression levels of miR-210 are higher in hypoxia-subjected cells relative to cells ectopically expressing miR-210 (miR-210 OE) (7.84 ± 0.780 vs 5.79 ± 0.562, statistically not significant). We thank the reviewer for highlighting this observation, as we believe that this observation is one of seminal focal aspects of the study. Continuing, in line of this thought, miR-210 levels are further synergistically enhanced, in cells ectopically expressing miR-210 (miR-210 OE) subjected to hypoxia (17.21 ± 1.846, Figure 1D bottom panel). This finding is pivotal in the context of the study and the reviewer’s assessment. We would like to further substantiate the significance of this observation (i.e. higher prevailing background levels of miR-210 across experimental groups), in the context of apoptotic cell death and GSK3β activation status, as follows.

  1. Generic cell death assay (LDH assay) and Apoptotic cell death assays (caspase-3 activity, DFF40 endonuclease, TUNEL) reveal that the underlying GSK3β (in)activation status governs the differential response that is observed in response to miR-210 overexpression in the context of hypoxia challenge. Case-in-point, when comparing miR-210 overexpressing hypoxia challenged cells on the wild-type GSK3β background versus miR-210 overexpressing hypoxia challenged cells on the ca-GSK3β-S9A (constitutively active GSK3β) background (the 4th experimental group vs the 8th experimental group depicted on graphical sets of data), there is a - 2.77-fold (0.2279 vs 0.6318) increase in LDH release (Figure 2A); 2.68-fold (0.2590 vs 0.6958) increase in caspase-3 activity (Figure 2F); 3.21-fold (0.2199 vs 0.7068) increase in DFF40 endonuclease activity (Figure 3D); 3.15-fold (0.2042 vs 0.6432) increase in TUNEL-positive cells (Figure 3A), despite the miR-210 expression levels being equitable between the groups (Supplementary Figures S1A, S1B, and S2B). In a stark contrast, when we compare ca-GSK3β-S9A expressing hypoxia challenged cells in the context of miR-210 overexpression background (the 6th experimental group vs the 8th experimental group depicted on graphical sets of data), there are no significant changes in LDH release (Figure 2A); caspase-3 activity (Figure 2F); DFF40 endonuclease activity (Figure 3D); TUNEL-positive cells (Figure 3A), despite the miR-210 expression levels being ~3.5-fold different between the experimental groups (Supplementary Figures S1A, S1B, and S2B). Thus, in hypoxia-challenged cells, under our experimental paradigm, cell death parameters show an increase in magnitude when GSK3β activity is increased, despite equitable miR-210 levels and in corollary analysis, cell death parameters exhibit no fluxes in magnitude when GSK3β activity is equitable, despite miR-210 levels being ~3.5-fold different. Thus, the prevailing disparity in miR-210 expression levels is not a confounding factor in the inferential analysis of this study.  

  1. In continuum with the latter aspects of the aforementioned discourse (last sentence, point 1), we do recognize, understand, appreciate, and agree with the reviewer’s comments regarding the scientific merit of testing the corollary hypothesis that - miR-210 downregulation / miR-210 inhibition could exacerbate or augment hypoxia-induced apoptotic cell death through enhanced GSK3β kinase activity. However, we have already exhaustively characterized, meticulously dissected, and explicitly delineated the role of GSK3β kinase inactivation in significantly mediating the miR-210 induced attenuation of hypoxia-induced apoptotic cell death. Taken together, it is our conviction that testing the corollary hypothesis may not add any greater scientific merit and weight to the findings and observations reported in this study.

Reviewer 2 Report

The manuscript by Gurdeep Marwarha et al. unveils that miR-210 attenuates hypoxia-induced apoptotic cell death in AC16 cardiomyocytes by augmenting the inhibitory phosphorylation of GSK3β at the Ser9 residue, consequently mitigating the kinase activity of GSK3β, without affecting its expression levels. The authors identify GSK3β as a distal, indirect molecular target of miR-210 and as a constituent of the miR-210 regulome.

The study is well constructed, original and very relevant for this scientific field. The experiments are well designed and the conclusions are strongly supported by the results.

The manuscript is well written and the results are clearly presented. The limitations of the study are highlighted in the discussion.

I have just a minor comment:

A graphical figure representing the downstream molecular targets of the miR-210-GSK3β signaling cross-talk could help understanding this complex pathway.

Author Response

The manuscript by Gurdeep Marwarha et al. unveils that miR-210 attenuates hypoxia-induced apoptotic cell death in AC16 cardiomyocytes by augmenting the inhibitory phosphorylation of GSK3β at the Ser9 residue, consequently mitigating the kinase activity of GSK3β, without affecting its expression levels. The authors identify GSK3β as a distal, indirect molecular target of miR-210 and as a constituent of the miR-210 regulome.

The study is well constructed, original and very relevant for this scientific field. The experiments are well designed and the conclusions are strongly supported by the results.

The manuscript is well written and the results are clearly presented. The limitations of the study are highlighted in the discussion.

I have just a minor comment:

A graphical figure representing the downstream molecular targets of the miR-210-GSK3β signaling cross-talk could help understanding this complex pathway.

We sincerely appreciate and thank the reviewer for the critical appraisal of our manuscript. We have now added a Graphical Figure / Summary Figure (Figure 7) that illustrates an integrated overview of the seminal findings reported in this study. In addition to the Graphical Figure, we have also added the following excerpt in the final paragraph of the Results section (Page 19, lines 274-282), as follows

“In an integrated overview, our data suggest that the inhibition of GSK3β kinase activity underlies the miR-210 elicited attenuation in the hypoxia-driven increase in the; OMM-insertion of BAX and BAK oligomers, the consequential release of Cytochrome C into the cytosol, the ensuing caspase-3 and DFF40 endonuclease activation leading to apoptotic cell death through the inhibition of GSK3β  The overall inferences drawn from our data, representing the findings and observations that the inhibition of GSK3β kinase activity is an indispensable molecular event and mediator of the miR-210 elicited mitigation of hypoxia-induced apoptotic cell death, are summated and presented as an illustrative schematic model in Figure 7.”

Round 2

Reviewer 1 Report

Authors have addressed the issues.